How many single-copy orthologous genes from whole genomes reveal deep gastropod relationships?

Chen Zeyuan 1 2 Chen.Z@snsb.de
Schrödl Michael 1 2 3
1 Mollusca, SNSB-Bavarian State Collection of Zoology , Munich, Bavaria , Germany
2 Department Biology II, Ludwig-Maximilians-Universität München , Munich, Bavaria , Germany
3 GeoBio-Center LMU , Munich, Bavaria , Germany
Steczkiewicz Kamil
Electronic publication date: 2022 Apr 18
Publication date: 2022
Volume: 10
Electronic Location ID: e13285
Received 2022 Jan 19; Accepted 2022 Mar 28
Copyright: © 2022 Chen and Schrödl
Copyright year: 2022
Copyright holder: Chen and Schrödl
License: This is an open access article distributed under the terms of the Creative Commons Attribution License, which permits unrestricted use, distribution, reproduction and adaptation in any medium and for any purpose provided that it is properly attributed. For attribution, the original author(s), title, publication source (PeerJ) and either DOI or URL of the article must be cited.
License URL: https://creativecommons.org/licenses/by/4.0/

Keywords: Phylogeny, Whole genomes, Gene tree, Mollusca, Patellogastropoda, Conflicting topologies, Orthogastropoda

Funding: European Union’s Horizon 2020 764840 This study has received funding from the European Union’s Horizon 2020 research and innovation programme under the Marie Skłodowska-Curie grant agreement No. 764840. The funders had no role in study design, data collection and analysis, decision to publish, or preparation of the manuscript.

==============================
The Gastropoda contains 80% of existing mollusks and is the most diverse animal class second only to the Insecta. However, the deep phylogeny of gastropods has been controversial for a long time. Especially the position of Patellogastropoda is a major uncertainty. Morphology and some mitochondria studies concluded that Patellogastropoda is likely to be sister to all other gastropods (Orthogastropoda hypothesis), while transcriptomic and other mitogenomic studies indicated that Patellogastropoda and Vetigastropoda are sister taxa (Psilogastropoda). With the release of high-quality genomes, orthologous genes can be better identified and serve as powerful candidates for phylogenetic analysis. The question is, given the current limitations on the taxon sampling side, how many markers are needed to provide robust results. Here, we identified single-copy orthologous genes (SOGs) from 14 gastropods species with whole genomes available which cover five main gastropod subclasses. We generated different datasets from 395 to 1610 SOGs by allowing species missing in different levels. We constructed gene trees of each SOG, and inferred species trees from different collections of gene trees. We found as the number of SOGs increased, the inferred topology changed from Patellogastropoda being sister to all other gastropods to Patellogastropoda being sister to Vetigastropoda + Neomphalina (Psilogastropoda s.l.), with considerable support. Our study thus rejects the Orthogastropoda concept showing that the selection of the representative species and use of sufficient informative sites greatly influence the analysis of deep gastropod phylogeny.

Introduction

Commonly known as snails and slugs, gastropods are the most common and relevant class within the phylum Mollusca. The earliest undisputed gastropods date from the Late Cambrian Period, around 500 million years ago, and now the total number of existing gastropod species is estimated to range from 63,000 to more than 100,000 (Bieler, 1992; Bouchet et al., 2017), accounting for 80% of known mollusks. Gastropods have a worldwide distribution, from the near Arctic and Antarctic zones to the tropics (Crame, 2013) and ubiquitous in the oceans. They have colonized even extreme environments, such as the deep sea (e.g., faunas associated with hydrothermal vents) (Govenar, Fisher & Shank, 2015) and deserts (Greve et al., 2017). Gastropods are highly diverse with regard to their body forms, shells, and functions, and in many ways, they affect human life. Some gastropod species (such as conch, abalone, limpets, and whelks) are used as food, the shells of some species were used as ornaments or in making jewellery (Frýda, 2013). A couple of gastropod species are intermediate hosts of parasites and transmit human or animal diseases, such as schistosomiasis (Fried & Huffman, 2017). Several land and freshwater snails and slugs are highly invasive and competitive species, feeding on crops and vegetables and causing threats to local ecology and economy, e.g., the notorious Spanish slug, Arion vulgaris Moquin-Tandon, 1855 (Zemanova, Knop & Heckel, 2017), golden apple snail, Pomacea canaliculata Lamarck, 1822, etc. (Global Invasive Species Database, 2021). All these characteristics: long evolutionary history, wide adaptability, high diversity, potential invasiveness, competitiveness, and high relevance for human lives, together with their long and rich fossil record (Frýda, 2013), make gastropods a unique animal group for evolutionary, ecological, and biogeographical investigations. Some of the most relevant questions concern the transition from ocean to freshwater to terrestrial lifestyle, the rapid adaptation in a new environment, the radiation evolution, the changes of shell morphology such as towards limpets or its reduction, internalization or loss (Solem, 2020).

The basis of all the above issues relies on resolving the gastropod evolutionary history, and there is great progress in subgroups, e.g., in Vetigastropoda (Cunha et al., 2019) or within Heterobranchia (Brenzinger, Schrödl & Kano, 2021; Kano et al., 2016). However, the early gastropod phylogeny, i.e., the relationship between major gastropod groups, is still controversial. The extant Gastropoda are divided into seven main lineages, namely Patellogastropoda, Neomphalina, Cocculiniformia, Vetigastropoda, Neritimorpha, Caenogastropoda, and Heterobranchia. Usually, Heterobranchia and Ceanogastropoda were recovered as sister groups (forming the clade Apogastropoda), and a close relationship of Neomphalina, Cocculiniformia, and Vetigastropoda has been supported (Lee et al., 2019). The main uncertainty refers to the position of Patellogastropoda (true limpets) as different data and analysis methods led to different conclusions (Zapata et al., 2014). Based on morphological data, Patellogastropoda has been recognized as the earliest-branching gastropod group, i.e., being the sister of all the remaining lineages together forming the clade Orthogastropoda (Golikov & Starobogatov, 1975; Haszprunar, 1988; Lindberg, 1988). The Orthogastropoda-Patellogastropoda topology has also been supported by Kocot et al. (2011) using 308 genes from transcriptome data. However, an extended sampling of transcriptomic markers favored a sister group relationship of Patellogastropoda and Vetigastropoda (combined clade Psilogastropoda), thus rejecting the monophyly of Orthogastropoda (Cunha & Giribet, 2019; Smith et al., 2011; Zapata et al., 2014). Early phylogenies based on complete mitochondrial genomes consistently recovered Patellogastropoda sister to Heterobranchia (Arquez, Colgan & Castro, 2014; Grande, Templado & Zardoya, 2008; Osca et al., 2014; Uribe et al., 2016), which was suspected to be a long branch artefact (Schrödl & Stöger, 2014; Stöger & Schrödl, 2013). More recently, a deep gastropod split into Patellogastropoda and Orthogastropoda was recovered by expanding the Patellogastropoda sampling (Uribe et al., 2019), however, this was still based on a limited number of relatively fast-evolving mitochondrial genes.

Ultimately, the increasing number of published high-quality gastropod whole nuclear genomes raised the possibility of identifying large numbers of putative orthologs across multiple species (Gomes-dos-Santos et al., 2019). Orthologs are defined as genes sharing a common ancestor by speciation, in contrast to paralogs, which are duplicated copies arising through polyploidization or duplications (Gogarten & Olendzenski, 1999; Sonnhammer & Koonin, 2002). Orthologous genes, if in a single copy status, are so-called single-copy orthologous genes (SOGs), which imply that they have kept this status since the species’ last common ancestor, when not considered rare cases, such as the differential loss of paralogs after whole-genome duplication or orthologous gene displacement (Creevey et al., 2011). SOGs thus hold great information potential for phylogenetic reconstruction, especially where universal markers are not able to generate strong phylogenetic hypotheses (Sang, 2002; Wu et al., 2006). Several newly published gastropod genome studies have tried to reconstruct gastropod phylogeny using hundreds of SOGs (Table 1); Lan et al. (2021) and Sun et al. (2020) recovered Psilogastropoda, while Chen et al. (2020) inferred Orthogastropoda (Table S1). Therefore, the conflict still remains when using SOGs in different taxon and gene sets with different species coverage. To explain the foundations leading to inconsistent topologies, we re-analyzed and expanded Chen et al.’s (2020) study. In our new analyses, establishing sets of SOGs we allowed species missing per SOG successively to a certain degree (0–20% of all species) and reconstructed the phylogenies. We found that as the number of species missing per SOG increases, the number of orthologous genes discovered also increases, and the final topology changed from Patellogastropoda-Orthogastropoda to Patellogastropoda sister to a combined clade of Vetigastropoda and Neomphalina (Psilogastropoda in a broader sense) with significantly increased support values.

Table 1 Summary of published gastropods genome and deep gastropods phylogeny based on corresponding genome studies using single-copy orthologous genes (SOGs).

Sub-class	Species	GenBank assembly accession	Assembly level	Topology	Used of SOGs	References	
H	Arion vulgaris	GCA_020796225.1	Chromosome	P,((H,C),(V,N))	223	Chen et al. (2020)	
H	Achatina immaculata	GCA_009760885.1	Chromosome	P,(H,C)	229	Liu et al. (2020)	
H	Achatina fulica	100647 (Gigadb)	Chromosome	(P,H)	675	Guo et al. (2019)	
H	Lymnaea stagnalis	GCA_900036025.1	Contig	–	–	BANG, 2016, Unpublished data	
H	Aplysia californica	GCA_000002075.2	Scaffold	–	–	Broad Institute, 2013, Unpublished data	
H	Biomphalaria glabrata	GCA_000457365.1	Scaffold	–	–	Adema et al. (2017)	
H	Candidula unifasciata	GCA_905116865.2	Scaffold	–	–	Chueca, Schell & Pfenninger (2021)	
H	Cepaea nemoralis	GCA_014155875.1	Scaffold	–	–	Saenko et al. (2021)	
H	Elysia chlorotica	GCA_003991915.1	Scaffold	–	–	Cai et al. (2019)	
H	Elysia marginata	GCA_019649035.1	Scaffold	–	–	Maeda et al. (2021)	
H	Limacina bulimoides	GCA_009866985.1	Scaffold	–	–	Choo et al. (2020)	
H	Physella acuta	GCA_004329575.1	Scaffold	–	–	Ebbs, Loker & Brant (2018)	
H	Plakobranchus ocellatus	GCA_019648995.1	Scaffold	–	–	Maeda et al. (2021)	
H	Radix Auricularia	GCA_002072015.1	Scaffold	–	–	Schell et al. (2017)	
C	Lautoconus ventricosus	GCA_018398815.1	Chromosome	–	–	Pardos-Blas et al. (2021)	
C	Pomacea canaliculata	GCA_004794335.1	Chromosome	(P,V),(H,C)	1,357*	Sun et al. (2019)	
C	Alviniconcha marisindica	GCA_018857735.1	Contig	–	–	HKUST, 2021, Unpublished data	
C	Batillaria attramentaria	GCA_018292915.1	Contig	–	–	Ewha Womans University, 2021, Unpublished data	
C	Colubraria reticulata	GCA_900004695.1	Contig	–	–	University of Konstanz, 2016, Unpublished data	
C	Marisa cornuarietis	GCA_004794655.1	Contig	–	–	Sun et al. (2019)	
C	Phymorhynchus buccinoides	GCA_017654935.1	Contig	–	–	BGI, 2021, Unpublished data	
C	Anentome Helena	GCA_009936545.1	Scaffold	–	–	IRIDION GENOMES, 2020, Unpublished data	
C	Babylonia areolate	GCA_011634625.1	Scaffold	–	–	Fisheries and Technical, Economic College, 2020, Unpublished data	
C	Conus betulinus	GCA_016801955.1	Chromosome	–	–	Peng et al. (2021)	
C	Conus consors	GCA_004193615.1	Scaffold	–	–	Andreson et al. (2019)	
C	Conus tribblei	GCA_001262575.1	Scaffold	–	–	Barghi et al. (2016)	
C	Lanistes nyassanus	GCA_004794575.1	Scaffold	–	–	Sun et al. (2019)	
C	Pomacea maculate	GCA_004794325.1	Scaffold	–	–	Sun et al. (2019)	
V	Steromphala cineraria	GCA_916613615.1	Chromosome	–	–	Wellcome Sanger Institute, 2021, Unpublished data	
V	Haliotis laevigata	GCA_008038995.1	Scaffold	–	–	Botwright et al. (2019)	
V	Haliotis rubra	GCA_003918875.1	Scaffold	–	–	Gan et al. (2019)	
V	Haliotis rufescens	GCA_003343065.1	Scaffold	–	–	Masonbrink et al. (2019)	
N	Gigantopelta aegis	GCA_016097555.1	Chromosome	(P,(V,N)),(H,C)	529	Lan et al. (2021)	
N	Chrysomallon squamiferum	GCA_012295275.1	Chromosome	(P,(V,N)),(H,C)	1,375*	Sun et al. (2020)	
N	Dracogyra subfuscus	GCA_016106625.1	Scaffold	–	–	Lan et al. (2021)	
P	Lottia gigantea	GCA_000327385.1	Scaffold	–	–	DOE Joint Genome Institute, 2012, Unpublished data	
Note:

Subclass H, C, V, N, P represents Heterobranchia, Caenogastropoda, Vetigastropoda, Neomphalina, and Patellogastropoda respectively. * Represents SOGs that can be found in at least 60% of taxa. Data without citations have been replaced by the data submitter and data publication date in NCBI. The species in bold are the species used to infer the phylogeny in this article.

Materials and Methods

Identification of gastropod SOGs sets

Species were selected as in the A. vulgaris genome study (Chen et al., 2020). These species include bivalve species, Argopecten purpuratus Lamarck, 1819 (Li et al., 2018) and Saccostrea glomerata Gould, 1850 (Powell et al., 2018) as the outgroup. The 14 gastropods cover five (of six) subclasses, specifically, 7 Heterobranchia: A. vulgaris, Achatina fulica Ferussac, 1821, Ac. immaculata Lamarck, 1822, Biomphalaria glabrata Say, 1818, Radix auricularia Linnaeus, 1758, Aplysia californica J. G. Cooper, 1863, Elysia chlorotica Gould, 1870, 4 Caenogastropoda: P. canaliculata, Marisa cornuarietis Linnaeus, 1758, Lanistes nyassanus Dohrn, 1865, Conus consors G. B. Sowerby I, 1833, 1 Vetigastropoda: Haliotis rufescens Swainson, 1822, 1 Neomphalina: Chrysomallon squamiferum C. Chen, Linse, Copley & Rogers, 2015 and 1 Patellogastropoda: Lottia gigantea G. B. Sowerby I, 1834 (Table 1). Protein coding genes of all species were downloaded and gene families were clustered using SonicParanoid v1.3.6 (Cosentino & Iwasaki, 2019) with default parameters. SonicParanoid is a graph-based orthology inference tool, given N input proteomes, SonicParanoid conducts all-vs-all protein alignment for N * (N − 1) between-proteome and N within-proteome pairs using MMseqs2 (Steinegger & Söding, 2017). Different SOGs data sets were extracted from the results by: (1) all 16 species, (2) allow one species missing per SOG, (3) allow two species missing per SOG, and (4) allow three species missing per SOG, resulting in a matrix with 395, 933, 1,331, 1,610 genes respectively (Fig. 1).

Figure 1 The four data sets used to infer gastropod relationships.

The first data set includes single-copy orthologous genes (SOGs) identified in all 16 species per SOG and results in 395 SOGs. 933, 1,331, 1,610 SOGs were identified while allowed 1–3 species missing per SOG respectively. The blue line represents SOGs identified by each species, and the black vertical line represents the missing of that gene in the corresponding species. The dots below represent the number of SOGs occupied by corresponding species under different degrees of species missing.

Phylogenetic analysis

For each SOGs set, amino acid sequences were aligned using MUSCLE v3.8.1551 with the default parameter (Edgar, 2004), and maximum likelihood (ML) gene trees were inferred using RAxML v8.2.12 with the parameter: “-f a -m PROTGAMMAAUTO -k -x 271828 -N 100 -p 31415” respectively (Stamatakis, 2014). The best-scoring ML tree of each gene was extracted separately and merged as input for ASTRAL v5.7.1 to estimate species trees with quartet support and posterior probabilities (PP) (Zhang et al., 2020).

For the subsequent testing of the species tree, we removed the species C. consors, which has a significant lower number of SOGs compared with other species. We used the same method to identify SOGs from all the remaining species. Protein sequences were then concatenated and aligned by MUSCLE v3.8.1551, and the maximum likelihood trees were inferred using IQ-TREE v2.0.3 with the parameter: “-m MFP -mtree -b 100”, and with an automatically selected best model (Table S1) (Minh et al., 2020).

Results

Our main goal was to explore the relationships between Patellogastropoda and other gastropods using SOGs depending on the number of species and genes involved. A total of 395 SOGs were first identified from all 16 species (including 14 gastropods species covering five main subclasses and two bivalves as the outgroup) (Table 1, Fig. 1). The topology of the species tree inferred is as described in Chen et al. (2020): Patellogastropoda is sister to all other gastropods with a low support (PP = 0.66), and the alternative topologies of Patellogastropoda being sister to Vetigastropoda + Neomphalina (PP = 0.27), Patellogastropoda sister to Heterobranchia + Caenogastropoda (PP = 0.07) even less likely (Fig. 2, Fig. S1). However, when we allowed one species missing per SOG, we got a total of 933 SOGs with each species covered by an average of 899 genes (Fig. 1). Surprisingly, the species trees change to Patellogastropoda as sister to Vetigastropoda + Neomphalina as dominant (PP = 0.83), the possibility of Patellogastropoda sister to all other gastropods highly decreased (PP = 0.17), and the posterior probabilities of Patellogastropoda sister to Heterobranchia + Ceanogastropoda decreased to 0.005 (Fig. 2, Fig. S2). With more species missing per SOG, the datasets of gene trees used for the estimation of species trees increased, and the posterior probabilities of Patellogastropoda sister to Vetigastropoda + Neomphalina gradually increased to 1 while the probabilities of the two alternative topologies decreased near to 0 (Fig. 2, Figs. S3–S4). Thus, inconsistent gastropod trees appear to be related to the size of the selected data set.

Figure 2 The gastropod relationships constructed with different data sets and the corresponding posterior probabilities of different topology.

H: Heterobranchia, C: Caenogastropoda, V: Vetigastropoda, N: Neomphalina, P: Patellogastropoda, B: Bivalve.

Furthermore, when examining the gene coverage of each species we found that the species C. consors is missing in most of the SOGs (Fig. 1). For example, when allowing one species missing per SOG, C. consors only showed 718 SOGs, which is 21% lower than the average level (mean = 911) (Fig. 1); allowing two to three species missing per SOG, this value increased from 30% (C. consors = 893, mean = 1,271) to 33% (C. consors = 1,007, mean = 1,505) (Fig. 1). We thus suspected the consideration of C. consors may greatly reduce the number of identified 1:1 orthologous genes, thereby affecting the inference of the phylogenetic tree. Our results support this hypothesis: excluding C. consors, we got a total of 847 SOGs (767,180 sites) among the rest of the 15 species, increasing SOG numbers 2.15 times. Then we used the 847 SOGs for phylogenetic reconstruction using a concatenation method and compared it with Chen et al. (2020) who used 233 SOGs (158,094 sites) identified in all 16 species. In contrast to Chen et al. (2020), Patellogastropoda clustered as sister to Vetigastropoda plus Neomphalina with a highly increased support (bootstrap value 89 compared to 48) (Fig. 3).

Figure 3 Gastropod phylogeny inferred from 847 single-copy orthologous genes (SOGs) identified from 15 species except C. consors.

Bootstrap support percentages are indicated for each internal branch. Branch length is marked with blue numbers.

Discussion

The lack of resolution and support in molecular phylogenies of early Paleozoic or even Precambrian groups, such as mollusks and gastropods, can be assigned to a plethora of potential problems regarding the sampling and selection of taxa, markers, and analyses (e.g., Schrödl & Stöger, 2014). One main issue is weak phylogenetic signal which may be eroded over time even in the most conservative genetic markers (Wagele et al., 2009). Selecting signal from noise by bioinformatic programs may benefit from expanding sets of suitable markers and taxa in, e.g., transcriptomic studies (Kocot et al., 2011; Zapata et al., 2014); however, without having delivered consistent and fully reliable hypotheses in the case of early gastropod evolution yet. High-quality whole genomes now offer the greatest possible set of genetic markers, with massive single-copy orthologs being the prime candidates for success (Creevey et al., 2011), but there is still a trade-off with the sampling of taxa, the representation and balance of major subgroups, and the quality and quantity of genomic data available per species. As a case study, here we focused on the role of number and coverage of taxa per SOGs using available quality whole genomes of gastropods.

Although the definition of SOGs is very clear, it is still challenging to correctly identify and optimize single copy (1-to-1) orthologs gene sets, especially on a large species scale. As expected, the numbers of detected SOGs shared by all gastropod species decreased as the number of quarried species increased, which highly influenced the results of the phylogenetic inference (Fig. 2). Without knowing the historic truth there is no direct way of assessing which (if any) of the controversial trees is correct or how many genes are necessary to recover it. However, using larger sets of SOGs revealed higher supported topologies than pursuing full species coverage with less SOGs: in our study, the gastropod phylogeny reconstructed by 80% species coverage of SOGs had the highest posterior probability of the whole tree (Fig. 2).

Moreover, the selection of species also needs to be cautious. In our case, the inclusion of C. consors led to a significant decrease in the number of shared SOGs and influenced the following phylogenetic inference. This might be caused by the fragmentation and incomplete assembly of C. consors (scaffold N50 = 1,128 bp) (Andreson et al., 2019), which in future studies can be replaced with the newly published chromosome level genome of cone snails, such as Conus betulinus Linnaeus, 1758 (Peng et al., 2021) and Lautoconus ventricosus Gmelin, 1791 (Pardos-Blas et al., 2021).

Furthermore, the currently available genomes constitute a far-from-optimal taxon set for resolving deep gastropod phylogeny: they are a small sample representing a large group, and those that are available come primarily from a rarefied pool (Sigwart et al., 2021). Such as L. gigantea, which is the only available Patellogastropoda, although it showed long-branch attraction and challenged deep gastropod mitogenomics (Stöger & Schrödl, 2013; Uribe et al., 2019). Whole genomes of Neritimorpha and Cocculiniformia are entirely missing and other major groups such as Neomphalina and Vetigastropoda rather than being represented by single or few members should be much more densely sampled, including the entire diversity of early branching subclades. As shown here, robust reconstruction of deep gastropod relationships will depend on using sufficient numbers of well-assembled whole genomes from a large and balanced taxon set.

The small and species fully covered SOGs data set used here and by Chen et al. (2020) just weakly supported Orthogastropoda. Our larger SOGs sets allowing 1–3 species missing per SOG, however, rejected Orthogastropoda and recovered Patellogastropoda as sister to a combined clade of Neomphalina and Vetigastropoda. This is in accordance with SOGs studies by Sun et al. (2020) (1,375 SOGs) and Lan et al. (2021) (529 SOGs) and supports large-scale transcriptomics by Cunha & Giribet (2019) who recovered Patellogastropoda sister to Vetigastropoda and called this clade the Psilogastropoda. Because they did not include Neomphalina into their transcriptomic study and established Psilogastropoda as the most inclusive clade containing Patellogastropoda and Vetigastropoda, our combined clade of Neomphalina plus Vetigastropoda as sister to Patellogastropoda supports and expands their Psilogastropoda hypothesis.

Conclusions

Striving for complete species covered SOGs in whole genomic studies lowered the total number of SOGs, while less stringent coverage largely expanded the set of SOGs, changed the topologies, and improved node supports. Only if using sufficient SOGs from an optimized taxon sampling, our study supports Patellogastropoda as a sister group to Vetigastropoda + Neomphalina, instead of Patellogastropoda as sister to all other gastropods. Future whole genome phylogenies are promising because of their unique potential to exploit the full genetic signal but should go for larger numbers of SOGs from more complete and better-balanced taxon sets and further explore the trade-offs of numbers, selection, and coverage of genes and species.

Supplemental Information

Supplemental Information 1 Gastropod phylogeny inferred from 395 single-copy orthologous genes (SOGs) using ASTRAL.

posterior probability (PP) is highlighted in red. Branch length is marked with blue numbers.

Click here for additional data file.

Supplemental Information 2 Gastropod phylogeny inferred from 933 single-copy orthologous genes (SOGs) using ASTRAL.

posterior probability (PP) is highlighted in red. Branch length is marked with blue numbers.

Click here for additional data file.

Supplemental Information 3 Gastropod phylogeny inferred from 1331 single-copy orthologous genes (SOGs) using ASTRAL.

posterior probability (PP) is highlighted in red. Branch length is marked with blue numbers.

Click here for additional data file.

Supplemental Information 4 Gastropod phylogeny inferred from 1610 single-copy orthologous genes (SOGs) using ASTRAL.

posterior probability (PP) is highlighted in red. Branch length is marked with blue numbers.

Click here for additional data file.

Supplemental Information 5 List of models sorted by BIC scores.

The best-fit model according to BIC: JTT+F+R5. AIC, w-AIC: Akaike information criterion scores and weights. AICc, w-AICc: Corrected AIC scores and weights. BIC, w-BIC: Bayesian information criterion scores and weights. Plus signs denote the 95% confidence sets. Minus signs denote significant exclusion.

Click here for additional data file.

Supplemental Information 6 The original data of gene family clusters.

Click here for additional data file.

Additional Information and Declarations

Competing Interests

Author Contributions

Data Availability

The authors declare that they have no competing interests.

Zeyuan Chen conceived and designed the experiments, performed the experiments, analyzed the data, prepared figures and/or tables, authored or reviewed drafts of the paper, and approved the final draft.

Michael Schrödl conceived and designed the experiments, authored or reviewed drafts of the paper, and approved the final draft.

The following information was supplied regarding data availability:

The raw data is available in the Supplemental File.

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
