# Peer review of "How many single-copy orthologous genes from whole genomes reveal deep gastropod relationships?"

_PeerJ, doi:10.7717/peerj.13285_

## Round 0.1 · original submission · Major Revisions

The reviewers were generally enthusiastic about the manuscript. Two major concerns are raised: one regarding picking the outgroup for phylogeny reconstruction which might affect the final result; and second regarding the dataset itself, specifically expanding it beyond the one used in previous work. Also, the reviewers requested more detailed comments on paralogy detection.

For me an interesting case would be, having the final phylogeny resolved, to check its congruence with single-gene trees including also paralogs to see which expansions are in par with the general statistical view on the gastropods radiation and which deviate from it (and possibly, why). Just a loose idea, may be worth exploring.

Since the reviewers suggested minor/major revision, I'm making it major to give the Authors more time for the detailed responses to reviews.

Reviewer 1 ·

Basic reporting

The references are current for the topic, and the figures are clear and of sufficient quality to enhance the article. The results match the hypothesis.

Experimental design

The intent and purpose of the paper is clear. The methods are outlined and replicable. This work is original.


I am not quite sure what the authors mean in the methods when they say “I species missing etc” and in figure 2, this needs to be explained in more detail as to what these species are exactly

Validity of the findings

While this paper presents an alternative hypothesis on the basal arrangements within the clade they are addressing, this fundamentally is a shift in the position of the basal taxa rather than a reorganisation of the branches per se. I have a concern in that the lack of use of similar outgroups may skew the result in the authors’ favour, and that a better method would be to include the same outgroups that the prior study that they seek to overturn had included.

Changing the included outgroups, in case omitting the Scaphopoda and the Cephalopoda (see Cunha and Giribet 2019. DOI 10.1098/rspb.2018.2776), may articfically support the authors hypothesis and indicates a possible selective bias in inclusivity of taxa to gain desires outcomes.

Additional comments

The first paragraph is a but "fluffy" and "waffly", and detracts form the quality of the over all paper.

Reviewer 2 ·

Basic reporting

See below

Experimental design

See below

Validity of the findings

See below

Additional comments

The early-branching events of the gastropods are indeed difficult to resolve, and is a long-standing question whose resolution would help malacologists in many aspects. If this study does not definitely solve the problem, it provides a nice correlation between the size of the dataset used and the topology obtained, and thus provide some guidelines for future studies. I have 2 major comments but, once considered, I think that this manuscript deserves publication in PeerJ:
1. The detection of the SOG is clearly one of the key-point of the methodological approach. Indeed, non-SOG genes would clearly biased the results, as the addition of more and more genes would also mean the additions of more non-SOG genes, and thus a bias in the resulting species trees. However, nothing is said about how the potential multi-copy and/or paralogues were detected. The approach proposed by SonicParanoid should be detailed a minimum. Also, it would have been interesting to check in the single-gene tree any trace of paralogy (i.e. well-established clades – e.g. caenogastropods, heterobranchs,… - that are not recovered monophyletic).
2. The dataset has been designed to match the dataset of Chen et al’s, but I think that updating the dataset with recently published genomes would be actually more interesting, even if it prevents a direct comparison with Chen et al’s tree. Furthermore, the inclusion of C. consors, as noted by the authors, clearly impact the size of the dataset, and I am not convinced that it is useful to maintain it in the study. Or if the goal is to test the impact of low-quality genome(s) on the size of the dataset, this should clearly be stated in the introduction as a goal of the study, and discussed accordingly (including the impact of other “low quality” genomes).
I am providing below a list of other minor points, plus more details on these 2 major comments.
L51: “gastropod types”: types is quite a vague concept… “species”, or “groups”?
L67-68: it is not very clear why the examples provided here are limited to quite narrow subgroups (in comparison with gastropods). Why not simply state that the phylogeny of shallow taxonomic levels has been tackled successfully in many groups, giving a few examples in each subclass?
L74: it seems to me that the close relationships of Neomphalina, Cocculiniformia and Vetigastropoda has been quite clearly demonstrated (eg. Lee et al 2019 - Incorporation of deep-sea and small-sized species provides new insights into gastropods phylogeny), and it does not seem to be controversy…
L83: see also Smith et al 2011 - Resolving the evolutionary relationships of molluscs with phylogenomic tools.
L115: why was it important to use the same dataset as Chen et al 2020? Why not using more recently published genomes? In particular, the chromosome-level genome of Lautoconus ventricosus instead of the poorly assembled Conus consors genome? The methodological approach could have been the same (except the direct comparison with the tree of Chen et al L178 – see below), but the final dataset could have been much better (in particular more SOG when missing species are allowed). From Fig 1 it is clear that Conus consors is the genome responsible for most of the loss of genes as more missing species are allowed.
L137-138: C. consors is finally removed from the dataset… Why not simply removing it from the manuscript? Or replacing it with Lautoconus ventricosus…
L139: it is not clear if the RAxML/ASTRAL analysis have been performed also without C. consors. From the results, it does not seem to be the case, but if the authors decide to keep C. consors in the dataset, it would have been interesting to perform the same type of tree reconstructions to make them comparable.
Results: even if the focus is on Patellogastropoda, it would have been nice to have a short statement about the rest of the tree, e.g. “the relationships of the other classes remain unchanged whatever the % of missing data (Caeno+Hetero, Veti+Neomph), only the Patellogastropoda position is affected”.
L158: the expression “With more species missing” used here and elsewhere can be misleading, and supposes that a whole species is missing, while it means that for each SOG, one species can be missing. I would write “with more species missing per locus”.
L178: I understand that the authors wanted to be able to compare their trees with Chen et al’s tree, but this is at the cost of using a more recent and complete dataset. The goal of the manuscript is to test the effect of missing data on the topology, and this is achieved with the four datasets used by the authors, and the comparison with Chen et al’s tree does not add a lot to the point.
L185: “High-quality”: some of the used genomes are not really of “high quality”, in particular C. consors.
L192-193: I fully agree, but again I am not convinced that it has been done properly here…
L200: “highest posterior probability”: do you mean the PP of the whole tree? Of the average PP of all the nodes?
L205-206: Lautoconus ventricosus is even better…
L221: it would be nice here to indicate the number of genes used by Sun et al and Lan et al.
Table 1, Column header: SOG instead of SCOG.

---

## Round 0.2 · accepted · Accept

Dear Authors,

The Reviewer accepted the revised version of the manuscript, congratulations!

Reviewer 2 ·

Basic reporting

No comment

Experimental design

No comment

Validity of the findings

No comment

Additional comments

No comment